# Functional Signatures in Non-Small-Cell Lung Cancer: A Systematic Review and Meta-Analysis of Sex-Based Differences in Transcriptomic Studies

**DOI:** 10.3390/cancers13010143

**Published:** 2021-01-05

**Authors:** Irene Pérez-Díez, Marta R. Hidalgo, Pablo Malmierca-Merlo, Zoraida Andreu, Sergio Romera-Giner, Rosa Farràs, María de la Iglesia-Vayá, Mariano Provencio, Atocha Romero, Francisco García-García

**Affiliations:** 1Bioinformatics and Biostatistics Unit, Principe Felipe Research Center (CIPF), 46012 Valencia, Spain; iperez@cipf.es (I.P.-D.); mhidalgo@cipf.es (M.R.H.); pmalmierca@cipf.es (P.M.-M.); zandreu@cipf.es (Z.A.); sergio.romera@atos.net (S.R.-G.); 2Biomedical Imaging Unit FISABIO-CIPF, Fundación para el Fomento de la Investigación Sanitaria y Biomédica de la Comunidad Valenciana, 46012 Valencia, Spain; delaiglesia_mar@gva.es; 3Atos Research Innovation (ARI), 28037 Madrid, Spain; 4Department of Oncogenic Signalling, Principe Felipe Research Center (CIPF), 46012 Valencia, Spain; rfarras@cipf.es; 5Medical Oncology Department, Hospital Universitario Puerta de Hierro-Majadahonda, 28222 Madrid, Spain; mariano.provencio@salud.madrid.org (M.P.); atocha.romero@salud.madrid.org (A.R.); 6Spanish National Bioinformatics Institute, ELIXIR-Spain (INB, ELIXIR-ES), C/Eduardo Primo Yúfera, 3, 46012 Valencia, Spain

**Keywords:** NSCLC, biomarkers, functional profiling, meta-analysis, sex characteristics

## Abstract

**Simple Summary:**

Differences in epidemiological and clinical patterns of lung adenocarcinoma have been described between male and female patients. Our study aimed to assess the molecular mechanisms underlying those differences through functional profiling and meta-analysis of lung adenocarcinoma expression datasets. We discovered an overrepresentation of terms related to acute inflammatory responses in female patients, suggesting an increase in certain cell populations, such as T CD8+ cells, and in interleukin production. We also identified purinergic signaling and lipid metabolism as relevant upregulated functional groups in female lung adenocarcinoma samples. Overall, we identified molecular processes and pathways that will aid the understanding of the differential clinical patterns described in lung adenocarcinoma in male and female patients and provide new pathways for the discovery of novel biomarkers and therapeutic targets.

**Abstract:**

While studies have established the existence of differences in the epidemiological and clinical patterns of lung adenocarcinoma between male and female patients, we know relatively little regarding the molecular mechanisms underlying such sex-based differences. In this study, we explore said differences through a meta-analysis of transcriptomic data. We performed a meta-analysis of the functional profiling of nine public datasets that included 1366 samples from Gene Expression Omnibus and The Cancer Genome Atlas databases. Meta-analysis results from data merged, normalized, and corrected for batch effect show an enrichment for Gene Ontology terms and Kyoto Encyclopedia of Genes and Genomes pathways related to the immune response, nucleic acid metabolism, and purinergic signaling. We discovered the overrepresentation of terms associated with the immune response, particularly with the acute inflammatory response, and purinergic signaling in female lung adenocarcinoma patients, which could influence reported clinical differences. Further evaluations of the identified differential biological processes and pathways could lead to the discovery of new biomarkers and therapeutic targets. Our findings also emphasize the relevance of sex-specific analyses in biomedicine, which represents a crucial aspect influencing biological variability in disease.

## 1. Introduction

Lung cancer is the most frequently diagnosed cancer and the leading cause of cancer-related death worldwide, representing 18.4% of all cancer deaths [1]. Exposure to tobacco, domestic biomass fuels, asbestos, and radon represent the most relevant lung cancer risk factors [1,2,3]; however, as has become evident from studies of other cancer types, sex-based differences (sexual dimorphisms) may also have significant relevance in lung cancer [1,4,5].

Lung cancer exhibits sex-based disparities in clinical characteristics and outcomes, with better survival observed in women [3,6,7]. While lung cancer incidence worldwide is higher in men, there exists an increasing trend in women that cannot be solely explained by tobacco consumption [1,2]. Furthermore, studies have reported sex-dependent differences in estrogen receptors and their impact on lung cancer [8,9,10]; however, conflicting results have attributed lung cancer susceptibility in women to genetic variants, hormonal factors, molecular abnormalities, and oncogenic viruses [3,11,12,13].

Adenocarcinoma represents the most frequent non-small cell lung cancer (NSCLC) subtype in both sexes [14], with a higher predominance in women compared to men (41% of cases in women versus 34% in men) [3,7,13]. Interestingly, Wheatley-Price et al. demonstrated a more pronounced survival rate difference between male and female lung adenocarcinoma patients when compared to other tumor types [7]. The molecular causes underlying such sex-biased patterns remain largely unknown, as limited efforts have been made for lung adenocarcinoma, with few studies considering this differential perspective [15,16,17,18]. These limitations can be partially addressed through meta-analysis, a robust methodology that combines information from related but independent studies to derive results with increased statistical power and precision [19,20]. As current treatment strategies do not cure most lung cancer patients, and invasive diagnostic techniques (e.g., via biopsies and bronchoscopies) often induce discomfort in patients, meta-analyses that improve our understanding of sex-specific molecular mechanisms in lung adenocarcinoma may facilitate the discovery of non-invasive prognostic and diagnostic biomarkers.

We performed a meta-analysis based on functional profiles of transcriptomic studies to explore the molecular mechanisms underlying sex-based differences in early-stage lung adenocarcinoma. We carried out exhaustive review and selection steps to guarantee the homogeneity of the selected studies and the subsequent comparison and integration of the data in the meta-analysis with an appreciation of this strategy’s specific limitations. After the systematic review, we retrieved and analyzed nine studies from Gene Expression Omnibus (GEO) [21] and The Cancer Genome Atlas (TCGA) [22], and then combined the results in a random-effects meta-analysis. This approach allowed the identification of functional alterations caused by lung adenocarcinoma in both male and female patients, comparing tumor samples with adjacent non-tumor tissue. In this study, we identified immune responses, purinergic signaling, and lipid metabolism as the main biological processes that display differences between male and female lung adenocarcinoma patients, with the acute immune response increased in female patients. Overall, our findings provide evidence that sex-based differences influence cancer biology and may impact response to treatment. Furthermore, underlying sex-based differences may contribute to the discovery of sex-specific prognostic and diagnostic biomarkers and the improvement of personalized therapies.

## 2. Results

We organized our findings into three sections: the first describes the studies evaluated and selected in the systematic review; the second section reports on the results of the bioinformatic analysis of each of these selected studies as follows: (i) exploratory analysis, (ii) differential expression, and (iii) functional characterization; while the third section presents the results of the differential functional profiling by sex.

### 2.1. Study Search and Selection

The systematic review identified 207 non-duplicated studies, of which 48.8% included both male and female patients (Appendix A). We applied inclusion and exclusion criteria (see Figure 1) to select a set of homogeneous and comparable studies to ensure the reliability of the subsequent analyses. To ensure study homogeneity (and in the hope of contributing to the early diagnosis of disease), we focused on those studies of early-stage disease and selected nine transcriptomic studies for further analysis (Table 1). The selected studies represented a population of 1366 early-stage samples (369 controls and 997 cases), of which 44% were from men, and 56% from women (Figure 2), with a median age of 65.54 years old. Table 1, Figure 2, and Appendix A contain further information regarding the selected studies and the clinicopathological characteristics of the study population.

### 2.2. Individual Analysis

As the normalized data derives from different platforms, we performed exploratory and processing steps for the data set to ensure the comparability and integration of subsequent analyses. The exploratory analysis found a lack of abnormal behavior except for three samples in the principal component analysis (PCA) and unsupervised clustering; therefore, we excluded the GSM47570 and GSM47578 samples in study GSE19188, and the GMS773784 sample in study GSE31210 from further analysis.

The differential expression results for each study demonstrated a large number of differentially expressed genes when comparing female lung adenocarcinoma samples to female control samples and male adenocarcinoma samples to male control samples (Appendix A). However, the evaluation of sex-based differences in lung adenocarcinoma patients provided a small number of significantly affected genes (see Appendix A), with no intersecting genes. 

We performed an individual functional enrichment analysis of Gene Ontology (GO) terms and Kyoto Encyclopedia of Genes and Genomes (KEGG) to identify the possible implications of these sex-specific differentially-expressed genes in pathways relevant to lung adenocarcinoma. The identified pathways revealed a diversity of significant results among datasets, which we have summarized in Table 2. When analyzing intersections, UpSet plots (Figure 3, equivalent to a Venn diagram) illustrate the degree of intersection between studies, demonstrating that most significant results are exclusive of each study. This data highlights the need for integrated strategies, such as meta-analyses, to increase the statistical power of any findings.

### 2.3. Meta-Analysis

We performed a functional meta-analysis for each of the 8672 GO functions and KEGG pathways, including every term found in at least two studies. Results with a false discovery rate (FDR) of <0.05 included 106 GO biological processes (BP), 3 GO molecular functions (MF), and 20 KEGG pathways, which were associated with 21 wider functional groups. We rejected potential bias on the significant results after the inspection of funnel plots; furthermore, sensitivity analyses failed to indicate alterations in the results due to the inclusion of any study. The results for the 129 significant GO terms and KEGG pathways are further detailed in Appendix A, including FDR, the log odds ratio (LOR), and its 95% confidence interval (CI), and the standard error (SE) of the LOR. In addition, Appendix A details the number of functions involved in each of the genes of this functional signature, and Appendix A displays the prognostic description of this set of significant functions.

#### 2.3.1. Upregulated Functions

We discovered that 43.88% of detected functions related to the immune response (Appendix A and Figure 4), which all displayed upregulation in female lung adenocarcinoma patients. This finding agrees with studies that report more robust innate and adaptive immune responses in women who, for example, present with more efficient antigen-presenting cells (APCs) than males [33,34]. The results provided evidence for the positive regulation of an acute inflammatory response, with CD8+ alpha-beta T cell differentiation and activation, B cell proliferation and activation, and an increase of interleukin (IL) biosynthesis, including IL-2, 6, 8, 10, and 17. Several immune-related signaling pathways also displayed differences between female and male lung adenocarcinoma patients. We discovered the upregulation of the MyD88-independent toll-like receptor signaling pathway, NIK/NF-kappa β signaling pathway, FC-epsilon receptor signaling pathway, B-cell receptor signaling pathway, Toll-like receptor signaling pathway, NOD-like receptor signaling pathway, and RIG-I-like receptor signaling pathways in female lung adenocarcinoma patients. Overall, these findings suggest the relevance of sex-based differences in immune responses to lung cancer, which may represent a significant contributor to the marked differences observed during disease progression in male and female patients. Furthermore, such findings may help to define novel therapeutic targets and may have important implications for immunotherapy.

We also uncovered evident sex-based differences in cell metabolism. Studies have shown the significant upregulation of lipid metabolism (ceramide and sphingolipid biosynthetic processes) in females and the higher utilization of carbohydrates by males [35,36]. Furthermore, lipid metabolism and signaling are widely accepted as major players in cancer biology [37]. “Metabolism- Nucleic acids metabolism and signaling” was the second most abundant functional group upregulated in female lung adenocarcinoma patients, comprising 8.63% of the altered functions. These GO terms and KEGG pathways are mainly related to purinergic signaling through G protein-coupled receptors and cytidine metabolism. Other functional groups upregulated in female lung adenocarcinoma patients include cell migration and homeostasis. Overall, further explorations of sex differences in metabolic pathways may provide new perspectives for treatment approaches with sex-specific effects.

#### 2.3.2. Downregulated Functions

It should be noted that 23.26% of the significant functions exhibited lower activity in female compared to male lung adenocarcinoma patients. Downregulated functional groups include those related to cell cycle progression, cell junctions, DNA repair and telomere protection, mitochondrial processes, neural development, post-translational changes, post-transcriptional changes, protein degradation, and transcription regulation (Appendix A and Figure 4). Taken together, the downregulated pathways in female patients suggest the existence of lower levels of oxidative stress compared to male patients [38], which may contribute to the existence of a less permissive tumorigenic environment.

### 2.4. Metafun-NSCLC Web Tool

The Metafun-NSCLC web tool (https://bioinfo.cipf.es/metafun-nsclc) contains information related to the nine studies and 1329 samples involved in this study. For each study, this resource includes fold-changes of genes and LOR of functions and pathways that users can explore to identify profiles of interest. 

We carried out a total of 8672 meta-analyses. For each of the 129 significant functions and pathways, Metafun-NSCLC depicts the global activation level for all studies and each study’s specific contribution using statistical indicators (LOR, CI, and *p*-value) and graphical representations by function as forest and funnel plots. This open resource hopes to contribute to data sharing between researchers, the elaboration of innovative studies, and the discovery of new findings.

Here, we also highlight the importance of including/reporting sex-related data in the results of clinical studies given their general importance in tumor risk, treatment response, and outcomes in lung adenocarcinoma and other cancers/disorders. The integration of sex-based differences in this manner has the potential to significantly impact the cancer biology field.

## 3. Discussion

Despite the profuse evidence for the influence of sex on rates and patterns of metastasis, the expression of prognostic biomarkers, and therapeutic responses in several cancer types [39,40], sex-based differences have not been consistently considered when studying cancer, designing therapies, or constructing clinical trials. Cases of NSCLC, including adenocarcinoma, exhibit differences in incidence, prevalence, and severity in female and male patients [1,3,7,41]. Elucidating the molecular basis for this sex-based differential impact will have clinical relevance, as this information can guide/improve both diagnosis and treatment.

Biomedical research generally underrepresents female patients, with sex-based differences rarely considered [42,43]. Our systematic review of transcriptomic studies revealed that only 48.8% of lung adenocarcinoma-related datasets considered both sexes, a figure similar (49%) to that reported by Woitowich et al. [43]. Sex-based differences impact disease biomarkers, drug responses, and treatment [42], and, therefore, sex must represent a critical component of experimental design. Added to this problem, we faced a lack of standardization among studies and detailed clinical information (i.e., mutations, smoking status, stages) when searching for suitable datasets. The consideration of Findable, Accessible, Interoperable, and Reusable (FAIR) data principles [44], a requisite for quality science, would ensure that generated data can be of further use throughout the scientific community.

To the best of our knowledge, only four studies have attempted to address the functional alterations caused by lung adenocarcinoma in both male and female patients—those by Shi et al. [16], which considered female patients, Araujo et al. [15], Yuan et al. [17], and Li et al. [18]. Shi et al. [16] integrated samples from two datasets for a differential expression analysis followed by a functional enrichment analysis, whereas Araujo et al. [15] independently processed six datasets and jointly analyzed their results. Yuan et al. [17] compared male and female patients with various cancer types (including lung adenocarcinoma) using the TCGA dataset, but did not include control samples in the statistical comparison. Li et al. [18] evaluated the differences in lung adenocarcinoma by focusing only on metabolic pathways. By including both male and female patients and controls in our gene expression comparison, in contrast with the analysis performed by Shi et al. [16] and Yuan et al. [17], we have effectively unveiled sex-based differences occurring in lung adenocarcinoma. Of note, 88% of the results reported by Yuan et al. [17] relate to the sex chromosome, but not necessarily due to cancer. Shi et al. [16] described the consequences of cancer development in female patients, but the authors failed to compare said effects with male patients. Although Araujo et al. [15] do not describe the statistical comparisons performed (and do not perform a statistical integration such as meta-analysis), the authors describe the results obtained for each dataset. Our study addressed sex-based differences in male and female lung adenocarcinoma patients through meta-analysis to address previous limitations and improve on those approaches employed by others. Despite certain supposed limitations to our approach (the presence of studies with different sample sizes and types of platforms), meta-analyses can integrate selected studies by eliminating inconsistency in individual studies, thereby increasing the statistical power, and highlighting robust disease-associated functions. We performed a meta-analysis using a random-effects model on Gene Set Enrichment Analysis (GSEA) results independently obtained from each study to evaluate the functions differentially altered between male and female lung adenocarcinoma patients. To make results comparable and reduce biases in the type of analysis used, we applied the same bioinformatics strategy from normalized expression matrices to GSEA results. This robust methodology integrates groups of data and provides results with higher statistical power and precision [19,20] and reveals findings that cannot be obtained through the intersection or addition of results in individual studies. Nevertheless, we selected only samples from early-stage lung adenocarcinoma patients to reduce variability and included smoking status in the differential expression linear model. While the inclusion of the mutational status of relevant genes to lung adenocarcinoma (e.g., EGFR) could have revealed important insight, the lack of this information in the majority of the studies and the resultant limited sample size hampered this aim. We would support the inclusion of this type of data as a requirement for the publication of new data to repositories, such as the GEO. 

The immune system plays a crucial role in the development of cancer [45], and several studies have reported sex-based differences in immune responses (reviewed by Klein and Flanagan [33]). Tumor cells evade the immune system using different strategies, including the modulation of antigen-presentation and the suppression of regulatory T cells. Therefore, sex differences in APCs and their downstream effector cells, among other components, may contribute to the sexual disparity observed in various aspects of cancer development and may significantly impact antitumor immunity and immunotherapy. Adult females generally present more robust innate and adaptive immune responses than males, as evidenced by increased phagocytic activity of neutrophils and macrophages, more efficient APCs, and differences in lymphocyte subsets (B cells, CD4+T cells, CD8+T cells) and cytokine production. Accordingly, our results demonstrate an enrichment of immune response-related terms in female lung adenocarcinoma patients, which agrees with the findings of Araujo et al. [15]. The analyzed functions suggest the positive regulation of CD8+ alpha-beta T cell activation and differentiation in female lung adenocarcinoma patients, which play an essential role in antitumor immunity [46,47]. Furthermore, Ye et al. [47] discovered a more abundant population of effector memory CD8+ T cells in female lung adenocarcinoma patients, which agrees with our results. A previous study described CD8+ lymphocyte levels as a prognostic biomarker in NSCLC [48], and specifically in lung adenocarcinoma [49], with a correlation between higher levels of CD8+ lymphocytes with higher survival rates and lower disease recurrence. Elevated levels of active CD8+ T cells in female lung adenocarcinoma patients could form part of the molecular mechanism underlying higher survival rates when compared to male lung adenocarcinoma patients. Activation of the Notch signaling pathway decreases CD8+ T lymphocyte activity in lung adenocarcinoma [50]; therefore, the downregulation of the Notch signaling pathway discovered in female lung adenocarcinoma patients could explain higher CD8+ T activity when compared to male lung adenocarcinoma patients.

Concerning the immune response, we also detected differences that supported the increased production of IL-2, which is known to stimulate T cell proliferation and the production of effector T cells, thereby amplifying the lymphocytic response [51]. Higher levels of IL-2 could also relate to increased activity of CD8+ T cells in female lung adenocarcinoma patients. Increased levels of IL-10 are also supported in female lung adenocarcinoma patients and, although IL-10 has anti-inflammatory and anti-immune activities [52,53], studies have suggested a dual role in cancer. In advanced lung adenocarcinoma, high expression of IL-10 receptor 1 correlates with worse prognosis [52], while IL-10 expression by T-regulatory cells inhibits apoptosis through Programmed death-ligand 1 inhibition [53]. Nevertheless, IL-10 correlates with better prognosis when expressed by CD8+ T cells in early-stage NSCLC [54], and it seems to activate the antitumor control of CD8+ T cells [55]. IL-2 and IL-10 could display increased activity in early-stage female patients, alongside a higher population of active CD8+ T cells than males, conferring women a survival advantage.

We also detected the positive regulation of IL-6 biosynthesis in female lung adenocarcinoma patients, with increased IL-6 levels correlating with worse prognosis in NSCLC patients in previous studies [56,57]. Network analysis in non-smoking female lung adenocarcinoma patients described IL-6 as one of the pathology’s central nodes [16], and these findings agree with our results, which provide evidence of the critical role of IL-6 in tumor progression in female lung adenocarcinoma patients. IL-8 and IL-17 exhibit increased production and biosynthesis in female lung adenocarcinoma patients, with said interleukins known to influence tumor growth and metastasis and correlate with worse prognosis [58,59,60].

Although altered immune responses can positively and negatively influence tumor progression, our findings have detected GO terms that point to an elevated acute immune response in female compared to male lung adenocarcinoma patients. Of note, sex-based immunological differences in lung adenocarcinoma might have an impact on immunotherapy response. Different studies have addressed the role of sex in immunotherapy [5,47,61,62] and established improved survival for female NSCLC patients. The discovered molecular pathways differentially activated between male and female lung adenocarcinoma patients may underlie phenotypic differences regarding immunotherapy response.

Sex-based differences in metabolism occur under physiological conditions and in the presence of cancer. Here, we detected an upregulation of purinergic signaling and nucleic acid metabolism in female lung adenocarcinoma patients, a finding not described in previous lung cancer studies. An NSLC-based study described an antitumor effect of the P2X4 receptor [63], which also exhibits sexual dimorphism in murine brain microglia [64]. Other P2 and A2 receptors play a role in NSCLC [63], but evidence of sex-based differences in receptor expression in human NSCLC patients has yet to be reported. Purinergic signaling and the role of purinergic receptors may also have relevance to innate and adaptive responses in different inflammatory and neurodegenerative diseases and several cancer types, including pancreatic ductal carcinoma (PDAC), hepatocellular, hepatobiliary carcinoma cells, and breast cancer [65,66,67,68,69]. Of note, studies have linked the upregulation of purinergic signaling with poor prognosis in PDAC [66]. We also discovered significant differences in lipid metabolism, with the positive regulation of ceramide and sphingolipid biosynthetic processes upregulated in female lung adenocarcinoma patients. The presence of lipids can promote tumorigenesis, while higher adipose tissue levels are associated with poorer outcomes in several cancers [37]. Thus, exploring the differential roles of purinergic signaling and lipid metabolism between male and female lung adenocarcinoma patients may represent an interesting proposition to improve sex-specific risk-stratification of patients, prevention, diagnosis, and treatment.

DNA damage and repair-related genes also presented sex-based differences in lung adenocarcinoma patients. In general, males present with a higher level of DNA damage, and females present with a lower DNA repair capacity [70], and, in agreement, we detected DNA repair and telomere protection as a downregulated functional group in female lung adenocarcinoma patients (with both mechanisms involved in tumor growth prevention [71]). Furthermore, we discovered the upregulation of DNA repair and an increase of DNA unwinding in male lung adenocarcinoma patients. DNA unwinding has emerged as a new target in cancer therapy with a primary focus on helicase inhibitors [72]. Besides, regarding DNA repair, the evaluation of poly (ADP-ribose) polymerase inhibitors in NSCLC cell lines has suggested potential therapeutic activity [73,74], and there may be value in exploring both treatment approaches in NSCLC patients, especially male patients. 

With these results in mind, we propose future studies focused on DNA repair and lipid/purinergic metabolism in female lung adenocarcinoma patients and the immune response in male lung adenocarcinoma patients in the hope of developing enhanced therapeutic strategies. 

Our study has characterized functional differences between the sexes in lung adenocarcinoma, shedding light on the functional basis behind this pathology in male and female patients. While our meta-analysis confirmed the conclusions of other studies, we also report previously undescribed alterations in biological processes that may broaden this field of study. Further knowledge regarding how those factors related to the functional mechanisms, described above, differentially impact male and female lung adenocarcinoma patients and may improve our understanding of the disease and improve treatment and diagnosis through biomarker identification. 

## 4. Materials and Methods

Bioinformatics and statistical analysis employed R software v.3.5.3 [75]. Appendix A details R packages and versions.

### 4.1. Study Search and Selection

Publicly available datasets were collected from GEO [21], ArrayExpress [76], and TCGA [22]. A systematic search of studies published in the period 2004–2018 was conducted in 2019 following the preferred reporting items for systematic reviews and meta-analyses (PRISMA) guidelines [32]. Two researchers involved in the study carried out the literature search, and the consistency of the review and selection procedures used was evaluated and confirmed. Several keywords were employed in the search, including lung adenocarcinoma (ADC), non-small-cell lung carcinoma (NSCLC), Homo sapiens, and excluding cell lines.

Eleven variables were considered for each study, including the clinical characteristics of the patients (e.g., sex and smoking habit) and experimental design (e.g., sample size and sample extraction source). The final inclusion criteria were:Sex, disease stage, and smoking habit variables registered;RNA extracted directly from human lung biopsies;Both normal and lung adenocarcinoma samples available;Patients who had not undergone treatment before biopsy;Sample size of > 3 for case and control groups in both sexes.

Finally, normalized gene expression data of six array NSCLC datasets (GSE10072, GSE19188, GSE31210, GSE32863, GSE63459, and GSE75037) and counts matrix of three RNA-seq NSCLC datasets (GSE81089, GSE87340, and TCGA-LUAD) were retrieved.

### 4.2. Individual Transcriptomics Analysis

Individual transcriptomics analysis consisted of three steps: pre-processing, differential expression analysis, and functional enrichment analysis (Figure 5a).

Data pre-processing included the standardization of the nomenclature of the clinical variables included in each study, normalization of RNA-seq counts matrix, and exploratory analysis. RNA-seq counts were pre-processed with the *edgeR* [77] R package using the trimmed mean of m-values (TMM) method [78]. We assessed the normalization methods performed by the original authors for each dataset, and log_2_ transformed the matrices when necessary. Annotation from probe set to Entrez identifiers from the National Center for Biotechnology Information [79] database and gene symbol was carried out with the *biomaRt* [80] R package. When dealing with duplicated probe-to-Entrez mappings, the median of their expression values was calculated. The exploratory analysis included unsupervised clustering and PCA to detect patterns of expression between samples and genes and the presence of batch effects in each study (Figure 5b).

Differential expression analyses were performed using the *limma* [81] R package. To detect differentially expressed genes in male and female lung adenocarcinoma patients, the following contrast was applied:

(ADC.W − Control.W) − (ADC.M − Control.M)

where ADC.W, Control.W, ADC.M and Control.M correspond to lung adenocarcinoma affected women, control women, lung adenocarcinoma affected men, and control men, respectively. Paired samples design was implemented, and tobacco consumption was included as a batch effect on the *limma* linear model to reduce its impact on data. *p*-values were calculated and corrected for FDR [82]. This comparison allows the detection of genes and functions altered by the disease and that have higher or lower activity in women when compared to men. Significant functions and genes were considered when FDR < 0.05.

Functional enrichment analyses were performed using the Gene Set Enrichment Analysis (GSEA) implemented in the *mdgsa* [83] R package. *p*-values were, again, corrected for FDR. For functional annotation, two functional databases were used: the KEGG Pathway database [84] and GO [85]. GO terms were analyzed and propagated independently for each GO ontology: BPs, MFs, and cellular components (CC). Those annotations excessively specific or generic were filtered out, keeping functions with blocks of annotations between 10 and 500. Intersections within groups were analyzed with UpSet plots [86] (Figure 5c).

### 4.3. Functional Meta-Analysis

Functional GSEA results were integrated into a functional meta-analysis [87] implemented with mdgsa and metafor [88] R packages. Meta-analysis was applied under the DerSimonian and Laird random-effects model [89], taking into account individual study heterogeneity. This model considers the variability of individual studies by increasing the weights of studies with less variability when computing meta-analysis results. Thus, the most robust functions between studies are highlighted.

A total of 6467 GO BP terms, 785 GO CC terms, 1207 GO MF terms, and 213 KEGG pathways were evaluated. *p*-values, FDR corrected *p*-values, LOR, and 95% CIs of the LOR were calculated for each evaluated function. Functions and pathways with FDR < 0.05 were considered significant, and both funnel and forest plots were computed for each (Figure 5d). These representations were checked to assess for possible biased results, where LOR represents the effect size of a function, and the SE of the LOR serves as a study precision measure [90]. Sensitivity analysis (leave-one-out cross-validation [88]) was conducted for each significant function to verify possible alterations in the results due to the inclusion of any study.

### 4.4. Metafun-NSCLC Web Tool

All data and results generated in the different steps of the meta-analysis are available in the Metafun-NSCLC web tool (https://bioinfo.cipf.es/metafun-nsclc), which is freely accessible to any user and allows the confirmation of the results described in this manuscript and the exploration of other results of interest.

The front-end was developed using the Angular Framework. All graphics used in this web resource have been implemented with Plot.ly except for the exploratory analysis cluster plot, which was generated with ggplot2 [91].

This easy-to-use resource is divided into five sections: (1) Summary of analysis results in each phase. Then, for each of the studies, the detailed results of the (2) exploratory analysis, (3) differential expression, and (4) functional profiling. The user can interact with the web tool through graphics and tables and search for specific information for a gene or function. Finally, Section (5) provides several indicators for the significant functions identified in the meta-analysis that inform whether they are more active in men or women. 

## 5. Conclusions

Sex-based molecular differences may influence the incidence and outcome of lung adenocarcinoma and, therefore, may have important clinical implications. We identified immune responses, purinergic signaling, and lipid-related processes as the main biological processes altered between male and female lung adenocarcinoma patients by a meta-analysis of transcriptomic datasets. Said processes exhibit increased activity in female lung adenocarcinoma patients, whereas other processes (such as DNA repair) are more active in male lung adenocarcinoma. Although further studies are required to verify and fully explore these findings, our results provide new clues to understand the molecular mechanisms of sex-based differences in lung adenocarcinoma patients and new perspectives regarding the identification of biomarkers and therapeutic targets.

## Figures and Tables

**Figure 1 cancers-13-00143-f001:**
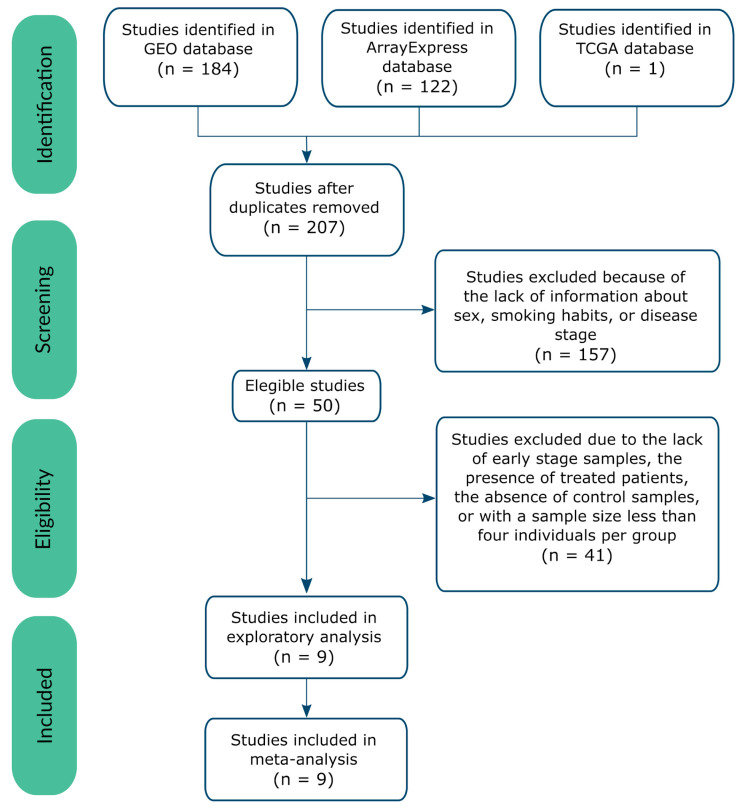
Flow of information through the different phases of the systematic review, following PRISMA Statement guidelines [32].

**Figure 2 cancers-13-00143-f002:**
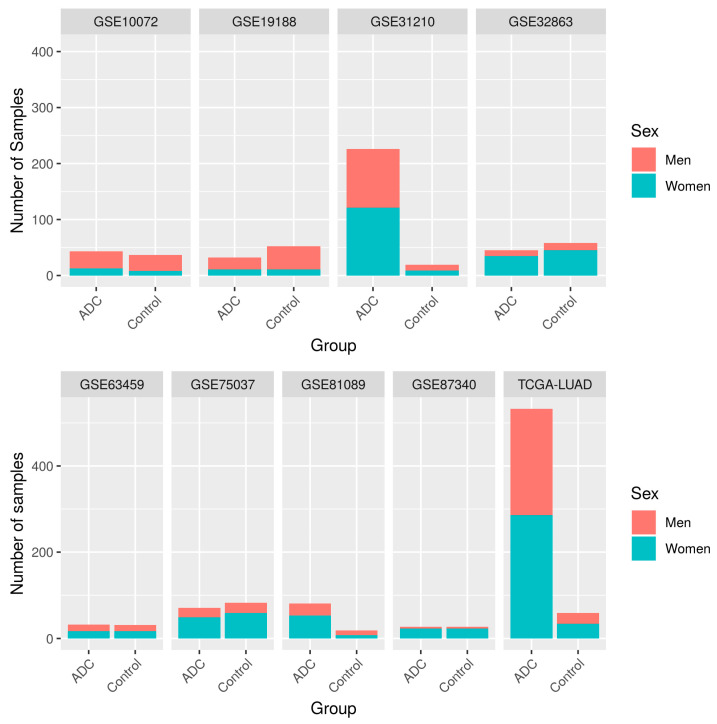
Number of samples per study, divided by sex and experimental group (ADC: lung adenocarcinoma samples).

**Figure 3 cancers-13-00143-f003:**
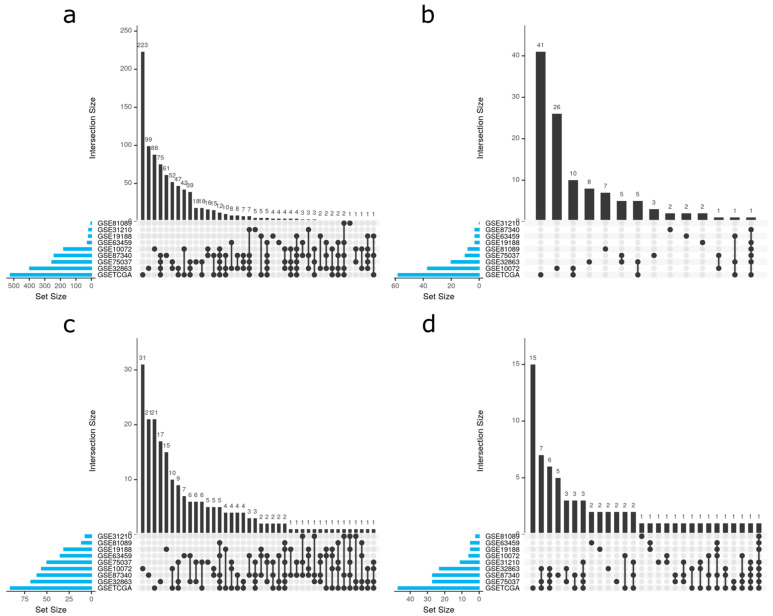
The intersection of significant functions between studies. UpSet plots for (**a**) Gene Ontology (GO) biological process, (**b**) GO molecular functions, (**c**) GO cellular components, and (**d**) Kyoto Encyclopedia of Genes and Genomes (KEGG) pathways. UpSet plots detailing the number of common elements among GO terms in our functional enrichment analysis. Horizontal bars indicate the number of significant elements in each study. The vertical bars indicate the common elements in the sets, indicated with dots under each bar. The single points represent the number of unique elements in each group.

**Figure 4 cancers-13-00143-f004:**
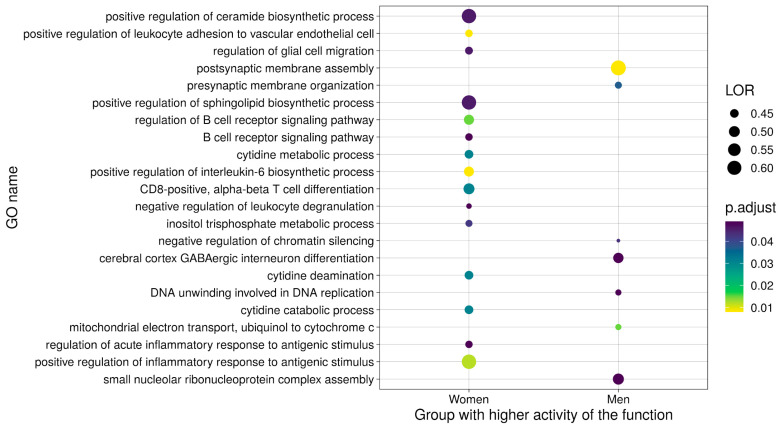
Summary dot plot of GO biological processes (BP) meta-analysis results. Only those significant terms with a log odds ratio (LOR) over 0.4 are shown.

**Figure 5 cancers-13-00143-f005:**
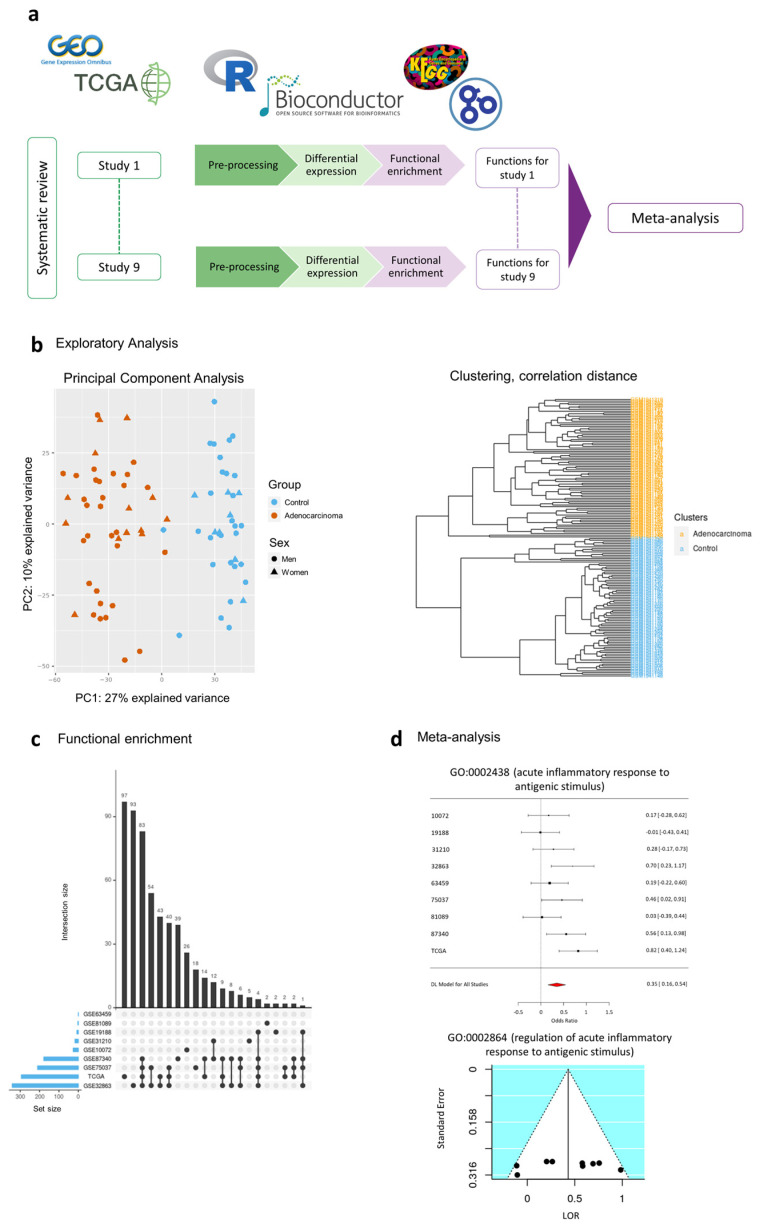
Workflow and analysis design. (**a**) Summary of the analysis design followed in this work, (**b**) PCA plot and HC analysis as an example of exploratory analysis (to explore possible batch effects and to assure expected data behavior) performed at the pre-processing stage to assess for the integrity of the data, (**c**) example of UpSet plot as an intersection analysis for functional enrichment analysis results, and (**d**) examples of forest and funnel plots to assess meta-analysis results.

**Table 1 cancers-13-00143-t001:** Studies selected after the systematic review.

Study	Platform	Publication
GSE10072	Affymetrix Human Genome U133A Array	[23]
GSE19188	Affymetrix Human Genome U133 Plus 2.0 Array	[24]
GSE31210	Affymetrix Human Genome U133 Plus 2.0 Array	[25,26]
GSE32863	Illumina HumanWG-6 v3.0 Expression BeadChip	[27]
GSE63459	Illumina HumanRef-8 v3.0 Expression BeadChip	[28]
GSE75037	Illumina HumanWG-6 v3.0 Expression BeadChip	[29]
GSE81089	Illumina HiSeq 2500	[30]
GSE87340	Illumina HiSeq 2000	[31]
TCGA	Illumina HiSeq 2000	[22]

**Table 2 cancers-13-00143-t002:** Summary of functional enrichment analysis results by Gene Ontology functions (BP: Biological Process, MF: Molecular Functions, CC: Cellular Component), and KEGG pathways. “Up” terms are overrepresented in female lung adenocarcinoma patients, while “Down” terms are overrepresented in male lung adenocarcinoma patients.

Study	Significant GO BP Terms			Significant GO MF Terms			Significant GO CC Terms			Significant KEGG Pathways		
	Up	Down	Total	Up	Down	Total	Up	Down	Total	Up	Down	Total
GSE10072	26	153	179	29	8	37	16	40	56	1	5	6
GSE19188	7	12	19	0	3	3	11	20	31	1	4	5
GSE31210	21	2	23	0	0	0	4	0	4	8	2	10
GSE32863	428	51	479	28	5	33	40	41	81	27	6	33
GSE63459	0	26	26	0	3	3	8	27	35	1	4	5
GSE75037	245	35	280	14	4	18	14	21	35	22	5	27
GSE81089	2	1	3	7	1	8	7	4	11	1	1	2
GSE87340	178	62	240	3	0	3	48	13	61	26	1	27
TCGA	294	228	522	28	30	58	21	70	91	30	17	47

## Data Availability

The data used for the analyses described in this work is publicly available at GEO (https://www.ncbi.nlm.nih.gov/geo/) and the Genomic Data Commons Data Portal (TCGA, https://portal.gdc.cancer.gov/). The accession numbers of the GEO datasets downloaded are: GSE10072, GSE19188, GSE31210, GSE32863, GSE63459, GSE75037, GSE81089, and GSE87340. Only the samples associated with the project TCGA-LUAD were downloaded from the TCGA.

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
