# Peer review of "Functional Signatures in Non-Small-Cell Lung Cancer: A Systematic Review and Meta-Analysis of Sex-Based Differences in Transcriptomic Studies"

_cancers, 2021, doi:10.3390/cancers13010143_

Round 1
Reviewer 1 Report
In this resubmission, the authors acknowledged the pan-cancer analysis of gender effects on patients’ molecular profiles performed by Yuan et al. and highlighted the methodological difference between these two studies. Yet, as mentioned in my previous comments, this work still lacks computational novelty and has insufficient evidence to support their claims. The authors should consider the following two aspects to increase the significance and credibility of their findings:
1) Immune-related genes on the sex chromosomes: Given many immune associated genes are on the sex chromosomes (PMID: 27546235, 22178198), please check what proportion of significant immune-related genes selected by your pipeline were 1) on the sex chromosomes and 2) overlapped with the genes included in Tables S1 and S4 from Yuan et al. Please contact the corresponding author if the gene lists are not publicly available. To facilitate data transparency, please provide the lists of genes that were overlapped with each of the 129 significant GO terms and KEGG pathways in your Supplementary Table S4.
2) Immune cell infiltration: Several groups reported that the types and amounts of tumor infiltrating immune cells were associated with patient prognosis in lung adenocarcinoma (PMID: 32081859, 32714316, 32346613, 30774636, 30275546). Your major findings were that immune-related genes (or pathways) from female patients were significantly upregulated compared with male patients, resulting in sex-based differences in immune responses to tumors. Please estimate the immune cell infiltration levels of each patient in LUAD by TIMER (PMID: 32442275, 29092952) or CIBERSORT (PMID: 25822800). Which types of immune cells are more abundant in female patients than in male patients? Were the GO terms or KEGG pathways related to the production of these immune cells selected by your pipeline?
Reviewer 2 Report
This is an interesting review and meta-analysis on the potential role of specific functional signatures to explain prognostic differences in NSCLC between female and male. To this end, the authors applied rigorous selection of published data to select the most relevant studies, and they applied state-of-the-art bioinformatics tools as far as I can judge. This study could set the stage for more in depth analyses in this area. Nevertheless, the are some points for consideration:
- The study claims that their results help to explain prognostic differences between male and female patients. They should try to confirm the prognostic effect of found transcriptional pathways in data from public databases linked to prognosis (e.g. TCGA & cBioPortal).
- The authors should provide more details on the “controls” that had been used. They mention “male control patients”, which is not clear to me. Usually adjacent benign lung tissue of the same resection specimen is used as matched control tissue. What control tissue was used from “other patients”? This must be clarified.
- Line 55: replace “aggressive” by “invasive”.
Round 2
Reviewer 1 Report
The authors made significant improvements in this revision. This work provides several interesting points in addition to Yuan et al. I am fully satisfied with their responses to my questions raised previously.
Reviewer 2 Report
The authors have adequately replied to my suggestions, and the quality of the manuscript has been improved.
This manuscript is a resubmission of an earlier submission. The following is a list of the peer review reports and author responses from that submission.
Round 1
Reviewer 1 Report
The authors sought to study the molecular differences in non-small-cell lung cancer between male and female patients. Through performing transcriptomic analysis of gene expression profiles publicly available in TCGA and GEO, they identified genes that 1) were differentially expressed between gender and 2) were enriched in some specific Gene Ontology terms and KEGG pathways such as immune-related signaling pathways. Lastly, the authors have deposited their results in a web-based tool, enabling users to study a gene or function of interest. From both methodological and application perspectives, however, this work is lack of novelties and suffers serious flaws. Similar attempts (PMID: 27165743, 32236130) have been previously published. While performing individual transcriptomic analysis between gender, the authors did not carefully control for other factors such as age. In addition, their meta-analysis did not account for technical differences between technologies (microarray vs. sequencing) and platforms (Illumina vs. Affymetrix). Put together, their discoveries were not supported by sufficient evidence and contributed very little to the field.
Reviewer 2 Report
Perez-Diez and colleagues present a meta-analysis of sex-specific signatures and differences in gene expression between males and females in NSC lung cancer (adenocarcinomas). The topic is very important- specifically there seems to be a growing body of literature pertaining to sex-specific treatable traits with differential responses to therapies. Whilst I am very enthusiastic about the topic, I have some questions about the approach and methods which diminishes my enthusiasm somewhat because it is difficult for me to interpret the data. These are my comments:
- I have concerns about some of the studies that are included in the meta-analysis in that they are not very good. For example, the studies with the smallest sample sizes are really not contributing anything to the analysis. Specifically, GSE63459, GSE10072, GSE19188 and GSED87340 are so small, I question their value. This becomes more of an issue because the largest of these cohorts has over 500 samples. I get the distinct feeling that the largest study is where almost all of the signal is emerging from, in other words, the analyses are not all that “meta”.
- There are also some issues with variance within the studies. Amongst these, I am most concerned by the microarray vs RNA-seq issue. Given that RNA-seq data is rich and TCGA is so large- the problem mentioned in #1 is all the more amplified. I believe that if you were to remove all the studies the signal that we are observing here would be much more reliable.
- It is impossible to work out how consistent and homogeneous the studies are. It would necessitate going through each of these studies, the point being that this information is not available in the manuscript. I would favour a table that presents this information with demographics, cancer staging, risk factors, and outcomes if known. For example, in one of the studies on patients without smoking history were included whereas the Robels study is famously of smokers only. There is a selection bias in both and one could question the validity of including this population in a meta-analysis of cohorts with multiple risk factors.
- There is no discussion on limitations- one which should be discussed is that there we don’t know if the associated findings are gender specific or are there some other factors that are driving differences in cancer mRNA expression. There are numerous factors that could result in these findings but the confounding data are currently shielded.
- I have found the figure somewhat difficult to follow. Figures 3A-D- what is the information that authors are trying to present? By takeaway from it is that RNA-sequencing identifies the most information with the most common elements and the other approaches are somewhat more variable. Perhaps I am missing something. Likewise Figure 5 seems difficult to follow too. What is the purpose of a PCA to reduce dimensionality of control vs Adenocarcinoma. We know these are markedly different from the original studies. Further why present the HC plot? Given the objective are to study differences in sex, these figures seem off topic. Incidentally, fig 5D is unreadable even at 250% zoom.
- There is not mention of quality control and outliers check for the array data (what was the quality of the signal?). Also the studies seem to be over a range of many years. During pre-processing, were the raw microarray normalized to compensate for the different array approaches used in the study? I think lack of concordance should be mentioned as a limitation. What about batch effect and its issues with variance; how that may affect their findings?
- How many people did the literature search?